# Optimising neonatal service provision for preterm babies born between 27 and 31 weeks gestation in England (OPTI-PREM), using national data, qualitative research and economic analysis: a study protocol

Thillagavathie Pillay,[1,2] Neena Modi,[3] Oliver Rivero-Arias,[4] Brad Manktelow,[5] Sarah E Seaton,[5] Natalie Armstrong,[5] Elizabeth S Draper,[5] Kelvin Dawson,[6] Alexis Paton,[5] Abdul Qader Tahir Ismail,[1] Miaoqing Yang,[4] Elaine M Boyle[5]

► Additional material for this paper are available online. To view these files, please visit the journal online (http://dx.doi.org/10.1136/bmjopen-2019-029421).

For numbered affiliations see end of article.

**Correspondence to**
Dr Thillagavathie Pillay;
tilly.pillay@nhs.net

## ABSTRACT

**Introduction** In England, for babies born at 23–26 weeks gestation, care in a neonatal intensive care unit (NICU) as opposed to a local neonatal unit (LNU) improves survival to discharge. This evidence is shaping neonatal health services. In contrast, there is no evidence to guide location of care for the next most vulnerable group (born at 27–31 weeks gestation) whose care is currently spread between 45 NICU and 84 LNU in England. This group represents 12% of preterm births in England and over onr-third of all neonatal unit care days. Compared with those born at 23–26 weeks gestation, they account for four times more admissions and twice as many National Health Service bed days/year.

**Methods** In this mixed-methods study, our primary objective is to assess, for babies born at 27–31 weeks gestation and admitted to a neonatal unit in England, whether care in an NICU vs an LNU impacts on survival and key morbidities (up to age 1 year), at each gestational age in weeks. Routinely recorded data extracted from real-time, point-of-care patient management systems held in the National Neonatal Research Database, Hospital Episode Statistics and Office for National Statistics, for January 2014 to December 2018, will be analysed. Secondary objectives are to assess (1) whether differences in care provided, rather than a focus on LNU/NICU designation, drives gestation-specific outcomes, (2) where care is most cost-effective and (3) what parents' and clinicians' perspectives are on place of care, and how these could guide clinical decision-making. Our findings will be used to develop recommendations, in collaboration with national bodies, to inform clinical practice, commissioning and policy-making. The project is supported by a parent advisory panel and a study steering committee.

**Ethics and dissemination** Research ethics approval has been obtained (IRAS 212304). Dissemination will be through publication of findings and development of recommendations for care.

**Trial registration number** NCT02994849 and ISRCTN74230187.

### Strengths and limitations of this study

► Scientific evidence from this study will be used to develop national recommendations for health service delivery for babies born between 27 and 31 weeks gestation in England.
► This will be guided by clinical outcomes, cost-effectiveness, parents' and staff perspectives.
► As a retrospective population-based observational cohort study, it is subject to selection bias in the assignment of location of birth of babies.
► Heterogeneity in the quality of care provided within and between local neonatal unit and neonatal intensive care unit, is likely, and will be addressed.
► Formal study-driven neurodevelopmental follow-up is not cost-effective in this large cohort, so routinely collected data will be used to investigate their outcomes.

## BACKGROUND AND RATIONALE

Specialised services for babies in England are delivered by neonatal units managed through Operational Delivery Networks.[1] These are geographical groupings of neonatal units working together in care pathways, and comprise units that are designated as neonatal intensive care units (NICUs), local neonatal units (LNUs) and special care baby units (SCBUs).[2 3] There were 15 operational delivery networks (ODNs) in England in 2018.[4] NICUs are located within centres that have specialist obstetric and fetomaternal medicine services; they have staff and resources to provide tertiary level care for babies of all gestational ages with a wide range and complexity of conditions. LNUs provide care for babies within their local catchment

area; they are able to provide emergency, short-term intensive care but are not resourced to provide long-term intensive care. SCBUs provide the lowest intensity of care usually for babies born >32 weeks gestation.

Caring for babies born at 23–26 weeks gestation in an NICU improves survival to discharge. This is now shaping policy for this category of babies.[5–7] However, there is no evidence to guide the location of care for the next most vulnerable group of babies born between 27+0 and 31+6 weeks gestation (hereafter called 'born at 27–31 weeks'). This is an important group, accounting for around four times the throughput in neonatal units compared with those born at 23–26 weeks gestation,[8] representing ~12% of all viable preterm babies born in England in 2013.[9]

Most ODNs have a defined gestational age cut-off, below which they aim to provide care in an NICU.[3 10] However, these criteria differ between networks and are often not adhered to. Currently, care pathways for babies born at 27–31 weeks gestation are undefined and their management is spread, often arbitrarily between NICU and LNU. While there is some evidence to suggest that morbidity profiles (based on common major morbidities such as necrotising enterocolitis, retinopathy of prematurity and bronchopulmonary dysplasia) for the total group of babies born at 27–31 weeks gestation[6] is similar between LNU and NICU, it remains unknown whether location of care and/or birth, makes a difference to gestation-specific outcomes. Our research will determine whether the type of neonatal unit in the hospital of birth influences outcomes for babies born at 27–31 weeks gestation, and if so, at which set point within this gestational age range care can equitably be provided in either LNU or NICU.

Presently, most pregnant women who want a hospital birth are advised to choose antenatal care and delivery in a hospital in their area,[11] which may have NICU, LNU or SCBU facilities. With the advent of Choose and Book,[12] this may be changing. Place of booking may also partly be determined by the anticipated degree of illness or complexity of care required for either mother or baby, if that can be predicted. However, for most women at the time of booking for pregnancy care, these risks are unknown and unpredictable. Women in threatened preterm labour at 27–31 weeks gestation will generally attend their hospital of booking for assessment or one closest to them at the time. Many will, therefore, deliver there, regardless of the type of neonatal care that can be offered. Babies that are inevitably born into an SCBU setting in this gestational age range are usually transferred ex-utero to an LNU or NICU for ongoing care. They may be returned to an SCBU from an NICU or LNU for 'step-down' hospital care if this is the unit closest to the mother's home, they are usually older than 32 weeks gestation and better, but not yet ready for discharge home. Generally, transfers that do occur between hospitals, whether for a mother before delivery or for her baby after birth, and are determined by care requirements, and by cot capacity and adequacy of neonatal nurse and medical staffing.

The total healthcare, social and education costs associated to prematurity at 27–31 weeks gestation during childhood has been estimated to be £607 million in 2016.[13] Survival in preterm babies has improved in recent years, and hospital costs are high. With neonatal intensive care averaging £1445 per day and high dependency care £925 per day[14] in 2017/2018, it is important to ensure that care for less sick preterm babies is not unnecessarily located within an NICU, and that sicker preterm babies are not undersupported in an LNU, potentially resulting in prolonged intensive care and greater short-term and long-term morbidity. Importantly, if key clinical outcomes do not significantly differ between LNU and NICU, then cost-effectiveness and family satisfaction should be the major drivers for how and where health services are delivered.

## METHODS AND ANALYSIS
### Objectives of OPTI-PREM workstreams
OPTI-PREM (Optimising neonatal service provision for preterm babies born between 27 and 31 weeks gestation in England) is a mixed-methods study comprising five workstreams. A summary of the study aims and objectives are described in figure 1 and expanded below.

### Patient and public involvement
Opti-Prem is supported by a parent advisory panel (figure 2) of representatives from BLISS, the acronym for the National Charity for babies born preterm or sick. The BLISS senior research engagement officer is part of the project's study steering committee. The chair of the parent advisory panel is a collaborator on the project.

Workstream 1: Clinical outcomes study

Workstream 1 aims to determine whether the type (designation) of unit (NICU vs LNU) at birth influences the primary outcome (mortality) and secondary outcomes (morbidities) in babies born at 27–31 weeks gestation.

Workstream 2: study of clinical care provided in different neonatal units

Variation exists in how neonatal units manage preterm babies of all gestational ages, even among units of the same designation.[15–18] In this workstream, we will address the secondary objective of identifying key differences in unit characteristics and clinical practices that could influence outcome. This will allow us to group neonatal units based on care provided, and to search for associations with gestation specific outcomes, irrespective of their designation.

Workstream 3: cost-effectiveness analysis

The health economics analysis investigates whether being born in settings with an NICU or LNU represent value for money within the National Health Service (NHS). A detailed cost analysis of the neonatal care received will be carried out. An incremental analysis of costs and the number of lives saved between different locations of birth will also be conducted.

**AIM**: The overarching aim of our research is to optimise neonatal service delivery for babies born at between $27^{+0}$ to $31^{+6}$ weeks gestation ('born at 27-31 weeks').

**OBJECTIVES**:

Primary objective:

1. For preterm babies born at 27-31 weeks population) and admitted into a neonatal unit for care: does care in a NICU (intervention), when compared to care in a LNU (comparator) result in improved gestation specific survival (primary outcome) and reduced major morbidity (secondary outcomes) up to 1 year of age?

   We will assess, within this primary objective, the impact of postnatal transfers between neonatal units (after day 1 of life) on gestation-specific survival and major morbidities up to 1 year of age.

Secondary objective:

2. For preterm babies born at 27-31 weeks and admitted into a neonatal unit for care:
   a) are there key differences in clinical care provided in LNUs vs NICUs, and are these associated with gestation-specific differences in outcomes;
   b) where is it most cost-effective to care for these babies from NHS perspective (LNU or NICU);
   c) what are parents' perspectives regarding place of care, and how can these guide decision making on place of care;
   d) what are LNU and NICU staff perspectives regarding place of care, and how can these be used to guide decision making on place of care.

The study findings will be used as a basis for discussions with the BAPM, the Neonatal Clinical Reference Group (CRG), Neonatal Specialist Commissioners, BLISS and relevant stakeholders such as Neonatal Nurses and Newborn Networks, in order to

- develop recommendations on the optimal, most cost-effective place of care for these babies, and
- promote implementation of these recommendations

**Figure 1** Overall aims and objectives for OPTI-PREM. BAPM, British Association of Perinatal Medicine; LNU, local neonatal unit; NICU, neonatal intensive care unit; NHS, National Health Service.

Workstream 4: ethnographic study with parents and clinicians

Using an ethnographic approach that accesses their lived experiences[19 20] of caring for these babies, this stream explores parents' and clinicians' experiences and perspectives regarding decisions about place of care.

Workstream 5: collaboration with the British Association of Perinatal Medicine (BAPM) to establish framework documents and recommendations on place of birth.

A working group will be established in collaboration with BAPM, which will engage with relevant stakeholders, the BLISS parent advisory panel and other support groups. Its purpose will be to establish a framework document and design recommendations for place of birth for babies born at each week of gestation between 27 and 31 weeks, based on the evidence from this study.

## Workstreams 1 and 3
### Population

All babies born at 27–31 weeks gestation between 1 January 2014 and 31 December 2018, in an English hospital with an LNU or NICU, whose records are captured within the National Neonatal Research Database (NNRD) will be included. With approximately 6000 babies per year in England meeting our criteria, the study is expected to include data on approximately 30 000 babies. Data on births and transfers to and from an SCBU will also be captured, to provide baseline information of place of birth, care and transfer across all neonatal units in England for this gestational age range.

Data sources:
1. Neonatal data: The NNRD will provide data on admissions to LNU or NICU in England. The NNRD is maintained by the NDAU and created from electronic health records.[21] It includes the information on

**Figure 2** Overview of OPTI-PREM workstreams. BAPM, British Association of Perinatal Medicine; LNU, local neonatal unit; NICU, neonatal intensive care unit; NNRD, National Neonatal Research Database; ODNs, operational delivery network;

characteristics of mothers and babies (eg, maternal pregnancy information, baby's birth weight, sex and gestational age), admission-related episodes (eg, reason for admission, admission temperature, length of stay and the number of episodes of care) and daily neonatal care of the babies after birth (eg, treatment, clinical procedures and morbidities).

2. Postneonatal data: To assess outcomes (survival and morbidities) to 365 days of age, the NNRD data will be linked to:
   i. Hospital Episode Statistics (HES): information all admissions, outpatient appointments and accident and emergency attendances at NHS hospitals in England.
   ii. Office for National Statistics (ONS): information on location, date and cause of death, obtained from death registration records.

Neonatal units will be allowed to assess the quality of their data entry onto electronic records, based on completion of data being fed back to neonatal units for surfactant replacement, nitric oxide and discharge home on oxygen.

Inclusion and exclusion criteria: Units will be offered an opt-out option at the start of the project. The

confidentiality of neonatal units who chose to opt out will be respected. A minimum dataset describing these units will only be reported if it is possible to maintain their anonymity in this process.

Clinical outcomes: The primary outcome for workstream 1 is death before discharge from a neonatal unit. We expect a within-unit mortality percentage of approximately 5% in this group. Information on deaths outside of neonatal care will be provided from ONS.

Secondary outcomes (eg, bronchopulmonary dysplasia, retinopathy of prematurity requiring laser treatment and worst cranial ultrasound abnormalities)[22] will be extracted from the daily care records in the NNRD. Data will be investigated for inconsistencies, out of range values and missing observations prior to the analysis of workstream 1. Special attention will be paid to morbidities that may be more relevant to this group of babies, such as growth during hospitalisation, blood stream infections, patent ductus arteriosus, time to full enteral feeds, time to breast feeding at initiation and breast feeding at discharge.

Statistical methods: In the absence of randomised treatment allocation, the difference in mean outcomes of babies born in either setting with an NICU or an LNU might be biassed if they are different in baseline characteristics, that is, our data sources are subject to selection bias. The analysis needs to account for the potential bias arising from the non-random allocation of babies to the two types of 'interventions'. We will tackle these issues in our analysis using matching and instrumental variables (IVs) methods. The same approaches will be followed by workstreams 1 and 3.

Matching to account for measured confounding: We will use matching to construct a balanced sample of babies who were born in NICU or LNU based on observed characteristics, assuming that these groups do not differ in relevant unobservable characteristics so that the differences in outcomes between them may be attributed to the location of birth. We will use propensity scores (PS), which are predicted probabilities of the relevant observed covariates that would be expected to be equal for both groups if the conditional distribution is independent of whether the baby is born in an NICU or LNU.[23 24] To compute the PS, a logistic model for place of birth will be fitted (LNU vs NICU). The independent variables will include a priori decided set of variables on the basis of expert and clinician input and/or variables associated with the type of unit at the place of birth. Based on previous literature[25 26] and the data availability in NNRD, we will evaluate a relatively large number of covariates for this model including: gestational age, birth weight, sex of the baby, mothers' ethnicity, use of antenatal steroids, 5 min Apgar score, deprivation, temperature at birth, maternal medical problems prior to this pregnancy and problems encountered during pregnancy. The selected variables are expected to influence either the place of birth or babies' outcomes, but none of them would be expected to be influenced by whether the baby is born in an NICU or LNU as they are defined at baseline prior

to or immediately after the birth. We believe that the model developed in the above steps will produce the PS given the observed set of covariates to produce balanced groups for the analyses; this means that babies who were born in NICUs and those who were born in LNUs with equal PS will have the same distributions of the observed covariates. After the PS is computed from the model, a matched sample will be created and the covariates will be compared between the groups in the matched sample to examine whether the balance has been achieved or not. If the sufficient balance is not achieved, the PS model will be modified and balance will be reassessed. We will investigate the range of estimated PS for both NICU and LNU babies to identify the region of common support. If there is no comparable NICU baby found in the LNU group, the analyses would produce biassed estimates by matching LNU babies with NICU babies with different characteristics. Therefore, observations not under common support should be dropped from the matching, but we will describe the characteristics of these babies in descriptive statistics. It is difficult to define the size of this matched sample at this stage. The sample size will be affected by how relevant it will be considered to the general population and will be reviewed by the study team. If required, other matching methods will also be applied, such as Mahalanobis distance matching and genetic matching.[27]

IVs to account for unmeasured confounding: Since matching is based on the assumption that the place of birth is independent of the potential outcomes conditional on the matching variables (also known as measured confounding). This requires that variables that affect both outcomes and the place of birth are observed and controlled in the matching algorithm. In our case, after matching the babies who were born in NICU or LNU on the basis of observed characteristics, it is possible that unobserved confounding variables still remain unbalanced between the two groups and the matching would result in biassed estimates of the effects of being born in NICU compared with LNU. Although we believe that the richness of our dataset makes the assumptions behind the matching plausible, we will complement our analyses using IVs.[28] We will explore two-stage least squared regression (2SLS) models with exclusion restrictions that help us to explain variation in the types of unit the babies were born that is independent of outcomes. In the first stage, we will estimate the probability of a baby's birth in NICU as a function of baby-level characteristics and IVs. The estimated probability of being born in an NICU will then be used in a second equation to estimate the effect of being born in an NICU on the outcomes of interest after controlling for a vector of observed baby-level characteristics. If the IVs are correct, the 2SLS method will generate unbiased estimates of the place of birth on babies' outcomes.

In general, a variable is considered to be an IV, if it is (1) associated with the type of unit at birth (NICU vs LNU), (2) does not have direct effect on the outcome (say mortality) and (3) is independent of unmeasured

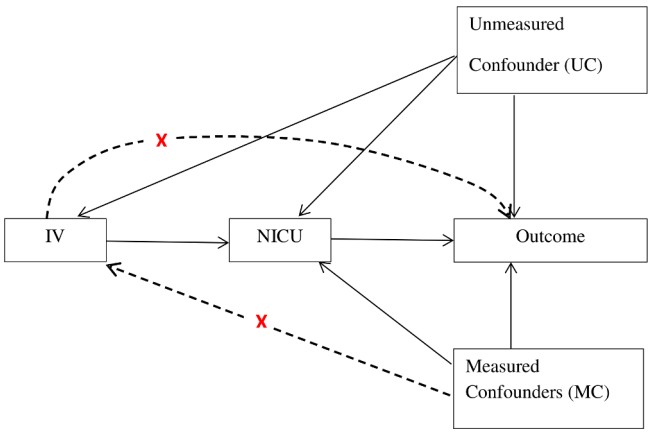

**Figure 3** Diagram of a valid instrumental variable (IV). NICU, neonatal intensive care unit.

confounders (UCs) but may be related to the measured confounders (MCs).[28 29] The assumed relation of (IV) with outcome (4), type of unit at birth (NICU vs LNU), UCs and MCs can be described by the graph in figure 3.[30]

A potential IV, used in previous studies in the USA,[26 30] is the difference of the distances of the mother residential postal code to the nearest NICU and LNU postal codes (excess distance). This has the potential to be an appropriate IV because of:

(1) Association with delivery in NICUs (refers to the arrow from IV to NICU in figure 3). Previous studies suggest that a high-risk mother is more likely to deliver in NICU if she is living close to an NICU hospital.[26]

(2) No direct effect on outcomes (refers to the crossed line from IV to outcome in figure 3). The assumption that the difference in travel times to either facility should not directly affect the outcomes except through the route of whether the baby is born in an NICU hospital,[29] (3) independence of unmeasured confounding factors (refers to the crossed line from IV to UC in figure 3). This is based on the assumption that mothers do not choose where to live based on distance to a high-level NICU, since preterm delivery is usually not expected before pregnancy. However, because hospitals with NICUs tend to locate in select areas (ie, cities), mothers who live here

may be different from those who do not, in terms of socio-economic characteristics. We attempt to address this issue by controlling for mothers' occupation and deprivation.

Analysing healthcare costs and cost-effectiveness outcomes: A cost analysis with a UK NHS perspective will be conducted. The cost of neonatal care received by preterm babies in NICU and LNU up to hospital discharge will be estimated. Standard care package in neonatal units will be costed by level of care, as categorised by the six Healthcare Resource Groups (HRG) as listed in table 1. HRGs are defined to group the use of healthcare resources of clinically similar treatments for costing purposes, and each of them is attached a unit cost that gives the average cost to the NHS of providing defined services to NHS patients in England. Unit costs used in our economic evaluation will be extracted from the NHS Reference Costs 2017/2018.

The costs of standard care provided by NICU or LNU, unit costs will be multiplied by the number of days using different levels of care. Since the standard packages costed by HRG codes do not include all major clinical activities, we will also cost the use of selected relevant procedures. These will include nitric oxide, surfactant replacement, total parental nutrition, ultrasound scanning, blood transfusion and palivizumab therapy. Cost data for these activities will be obtained from literature reviews and extracted from selected hospital finance departments as needed. The cost of surgical procedures are included in the standard care package by the level of care defined by HRG codes and will not be costed separately. Key components of healthcare resource use will be presented separately from associated costs as currently recommended.

Original neonatal care information will be linked to HES to extend the time horizon of the economic evaluation to 1 year of corrected age. A comprehensive cost analysis using the methodology above will be conducted to identify whether patterns of service use of hospital admissions, outpatient visits and accident and emergency services differs across neonatal strategies at 1-year-follow-up.

The number of lives saved will be the health outcome measure that will be used in the economic evaluation.

| HRG codes | Types of care | National average unit cost (2017–2018) |
|---|---|---|
| XA01Z | Neonatal critical care, intensive care | £1445 |
| XA02Z | Neonatal critical care, high dependency care | £925 |
| XA03Z | Neonatal critical care, special care, carer not resident alongside baby | £605 |
| XA04Z | Neonatal critical care, special care, carer resident at cot side and caring for baby | £435 |
| XA05Z | Neonatal critical care, normal care | £441 |
| XA06Z | Neonatal critical care, transportation | £1159 |

**Table 1** Reference cost for neonatal critical care—year 2017/2018—NHS England[14]

HRG, Healthcare Resource Group; NHS, National Health Service.

Deaths occurring after discharge from the neonatal unit will be informed by the ONS linked. An incremental analysis of costs and health outcomes between NICUs and LNUs will be conducted and synthesised using the net-benefit framework. Current thresholds of willingness to pay as recommended by the National Institute for Health and Care Excellence (NICE) will be used to determine value for money.

We will follow current guidance for methods of technology appraisal to present and report the results of the economic analysis.[31–33]

## Workstream 2

Adjusting for heterogeneity within the types of units studied: Using NNRD data and augmented with novel evaluation of the quality of care provided in neonatal units through a PhD project, other parameters such as associations with volume and calibre of work undertaken, medical and nursing staffing, and comparisons against national neonatal auditable standards (using variables such as normal temperature measured within the first hour of life, receipt of breast milk on discharge, parental consultation within the first 24 hours, parent presence on ward round, blood stream infections,among others), will be explored in workstream 2. This information, along with information about the population of babies which are cared for at their unit (but not including outcomes), will be used to undertake a cluster analysis. This analysis will provide groups of units which are similar in terms of their characteristics and the care they provide, irrespective of their designation. The identified variables will be assessed in two ways:

1. They will be incorporated into workstream 1 as confounding variables to analyse potential heterogeneity between units, and to investigate to what extent they help to explain this.
2. They will be used independently, in isolation and conjunction, to investigate associations with gestation-specific outcomes.

## Workstream 4

Understanding parent and clinician perspectives on decisions about place of care: Observations will explore: (1) factors that parents think should guide decision-making about place of care for babies, and how this happens in practice, (2) clinicians' perspectives and practices around decision-making about place of care, (3) the impact on parents and families of this decision, and subsequent changes in care location and (4) how parents can best be supported.

Periods of observation within neonatal units will be completed, along with interviews with both clinicians and parents. An experienced qualitative researcher will observe relevant discussions and interactions that take place between parents and clinicians regarding place of care. Observations will be guided by an observation framework, developed through discussions with the project team and project parent panel. The observer will take written notes unobtrusively and will then debrief these either alone or with another member of the team. These debrief sessions will be audio recorded and transcribed.

Alongside observations, interviews will be conducted with parents and clinicians on their perspectives and experiences of decisions regarding place of care. We will also interview a separate group of parents about their retrospective experiences of having had a baby receive neonatal care in the past 12 months. Interview topic guides will be developed and piloted through discussions with the project team and parent advisory panel. Interviews will be audio recorded and transcribed verbatim.

Participants, who are willing and able to give informed consent, and are caring (or did care) for a baby born between 27 and 31 weeks gestation, will be included. Participation will be voluntary and it will be made clear to parents in particular that declining will not compromise the care they or their babies receive. Informed consent for participation will be obtained (written for formal interviews and verbal for observation).

We will conduct up to 40 interviews with parents and clinicians. Twenty interviews will be 'retrospective' interviews with parents whose babies had been discharged in the last 12 months. These will be selected from the 15 operational delivery neonatal networks in England. This will be undertaken through an open national advertisement for parents to participate, through the national parent charity for sick and preterm babies, BLISS. We will attempt to recruit a diverse sample of parents with babies born at different gestations between 27 and 31 weeks, parents for whom this is a first or subsequent pregnancy, and parents of babies on different care pathways (eg, moving from LNU to NICU and vice versa). We will attempt to ensure diversity of participants in relation to maternal age, ethnicity, socioeconomic status and educational background. A further 20 will be 'real-time' interviews with clinicians and parents caring for a baby currently receiving neonatal care. These will be conducted in two operational delivery networks, situated in the West and East Midlands, the two neonatal networks closest to the study centre, for practical reasons. For these interviews, we will also include clinicians of different ethnic backgrounds, ages and sexes.

The qualitative data will be analysed using the constant comparative method[34] assisted by NVivo software. A coding scheme developed through detailed engagement with the data will be used to process the dataset systematically by assigning each section of text to a category, according to the category specifications. Parent panel members will be invited to comment on the face validity of the analysis of qualitative data.

## DISSEMINATION

A. Written information materials have been developed for both parents and clinicians invited to participate in the ethnographic study.

B. Presentation and publication: This will include presentation of research findings at national and international conferences, and publication in peer-reviewed journals.

C. Development of recommendations for care of the neonate born at 27–31 weeks gestation: A working group will be formed with BAPM and BLISS, to develop position statements and recommendations regarding the most appropriate place of care for babies born at this gestation in England. This will be led by investigators from the study and will incorporate nursing and medical professionals with relevant expertise, and representatives from the project's parent advisory panel.

D. Consultation with healthcare professionals and other stakeholders: Framework documents and recommendations will be processed through BAPM. This includes consultation with members and stakeholders (managerial stakeholders including Commissioners, Networks, Trusts, Governmental/Regulatory stakeholders including the Department of Health, Public Health England, National Commissioning Board, Care Quality Commission, NICE and educational stakeholders including Royal Colleges and professional societies).

E. Publication and dissemination of recommendations: Recommendations will be published via BAPM and RCPCH. Further dissemination will occur via the neonatal CRG to inform evidence-based commissioning of neonatal services, and BLISS to inform written information for parents, parent counselling and support services.

**Author affiliations**
[1]Royal Wolverhampton Hospitals NHS Trust, Wolverhampton, UK
[2]School of Medicine and Clinical Practice, University of Wolverhampton Faculty of Science and Engineering, Wolverhampton, UK
[3]Department of Neonatal Medicine, Imperial College London, London, UK
[4]Nuffield Department of Population Health, University of Oxford, Oxford, UK
[5]Department of Health Sciences, University of Leicester, Leicester, UK
[6]Parent Representative, BLISS National Charity for Babies Born Premature or Sick, London, UK

**Contributors** TP developed the idea for the project and developed the protocol in conjunction with EMB, and the Opti-Prem collaborating team. NM contributed to protocol development and provided guidance regarding NNRD data utilisation. TP, EMB and AQTI developed the concepts around workstream 2. TP and EMB are supervising AQTI's PhD project around this workstream. BM and SES developed the statistical methodology for the project. OR-A contributed to the protocol development, the statistical methodology for workstreams 2 and 3, and the study design of the cost-effectiveness analysis in workstream 3. MY contributed to the development of the statistical methodology for workstreams 2 and 3 and contributed to the study design of the economic analysis of workstream 3. NA and AP developed the ethnographic element (workstream 4) of the project. ESD provided epidemiological and overall support for the protocol. KD is the lead for the BLISS parent panel and has provided support and counsel at all stages of the project thus far.

**Funding** This work is supported by the National Institute for Health Research, Health Services and Delivery Research Stream, project number 15/70/104 CRN accrual was approved by the NIHR for the period (1 August 2017 to 31 August 2018).

**Competing interests** None declared.

**Ethics approval** Research ethics approval has been obtained through the national Integrated Research Application System (IRAS, reference number 212 304 and research ethics committee reference number 17/NE/0800). For workstreams 1, 2, 3 and 5, a proportionate review was undertaken. For workstream 4, research ethics approval was obtained together with R&D approval from the individual Trusts at which the interviews and observations will be conducted.

**Provenance and peer review** Not commissioned; externally peer reviewed.

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
