## [Reviewer comments · BMJ Open]

ARTICLE DETAILS

TITLE (PROVISIONAL)	Optimising neonatal service provision for preterm babies born between 27 and 31 weeks gestation in England (OPTI-PREM), using national data, qualitative research and economic analysis: a study protocol
AUTHORS	Pillay, Thillagavathie; Modi, Neena; Rivero-Arias, Oliver; Manktelow, Brad; Seaton, Sarah; Armstrong, Natalie; Draper, Elizabeth; Dawson, Kelvin; Paton, Alexis; Ismail, Abdul Qader; Yang, Miaoqing; Boyle, EM

VERSION 1 – REVIEW

REVIEWER	Véronique PIERRAT INSERM UMR 1153 Obstetrical, Perinatal and Pediatric Epidemiology Research Team (EPOPé). Center for Epidemiology and Statistics Sorbonne Paris Cité (CRESS). DHU Risks in Pregnancy Paris Descartes University. France
REVIEW RETURNED	13-Feb-2019

GENERAL COMMENTS	This study protocol is a very interesting one, investigating at the level of a population, a key question in neonatal care: the provision of neonatal care for preterm babies born between 27 and 31 weeks gestation. The global aim of this research is to optimise neonatal service deliveries for these babies, representing a large proportion of all viable preterm babies in industrialized countries, and an even larger proportion of children admitted to NICU or LNU. The main strength of the protocol is to combine different sources of information using national data, qualitative research and economic analysis. Other strengths are analyses strategies to account for measured and unmeasured confounders, the objectives of workstream 2 to better define NICU and LNU, and the deep involvement of parents and stakeholders in the project with an already planned dissemination strategy. I suggest minor clarifications: - Would it be suitable to give a few more information on the organisation of care in England during the study period? Do SCBU have a role in the care for babies born at 27-31 weeks? In particular, do SCBU admit babies transferred from NICU or LNU after 32 weeks? If yes, how this will be taken into account in the results?- It could also be interesting to know the number of networks available in England during the study period and consequently how the 2 networks will be chosen for parents and clinicians interviews (page 12, line 34).
--

	- Could the authors precise if a special attention will be paid to morbidities or care that could be as important for babies born at 27-31 weeks than BPD, ROP or cranial abnormalities, such as blood stream infections, growth during hospitalisation, breastfeeding initiation and/or at discharge for example? (Page 7, line 13 and 34) - "Inclusion and exclusion criteria: units will be offered an opt-out option" (Page 7, line 34). Is it planned to have a minimal dataset to describe units choosing an opt-out option? - Is it possible to give a minimal list of "national neonatal auditable standards" that will be taken into account and considered as useful for the care of babies born between 27-31 weeks? (Page 11, line26) Personal suggestion: The results expected with this study protocol are very important for neonatal service provision because of the quality of the overall protocol but also because of the target population. It is probable that information about other outcomes than mortality and severe neonatal morbidities (BDP, ROP, severe cerebral lesions) will be available through the NNRD and used by the group. The main outcomes that will be reported in this study are fundamental with regards to its objectives, but they are not very frequent in this population, especially after 29 weeks of gestation. It could thus be interesting to precise if a special focus will be made on morbidities that could also be useful to look at in the context of care for babies born at 27-31 weeks. I think that this could reinforce the originality and impact of the study. It might be useful to present it in more detail.
--	--

REVIEWER	Katie Mckinnon Neonatal Clinical Research Fellow University College London Hospital UK
REVIEW RETURNED	02-Apr-2019

GENERAL COMMENTS	A very interesting and important study, looking at this under-investigated problem from a variety of perspectives. A few small points follow. Page 4, paragraph starting line 52: "...most pregnant women book for antenatal care and delivery in the hospital closest to their home" I think this sentence would benefit from a reference. Since the advent of choose-and-book, and more information about maternity services available to the public, many women do choose to deliver in a hospital other than their local. This is mentioned in the section on Instrumental Variables (pages 9-10), but perhaps this could be expanded on as a factor affecting location of care? It also leads to an increased need for postnatal transfers to other hospitals and so on. Information about transfers generally, and the effect of this on care could also be expanded on. It is listed as part of the primary objective (page 22, line 26), but isn't really discussed elsewhere within the discussion of the methods as a factor affecting outcome. Page 5, line 43: Should read "...from BLISS, the charity for babies..."
--

	Page 8, line 40 onwards: Creating a matched sample is a logical step, however it is obviously difficult to assess how large this sample size will be at this stage in the project. As such it may be beneficial to mention that the size of this matched sample may affect how relevant it is to the population generally. Page 11, line 9: Should read "...framework. Current thresholds of willingness..."
--	--

VERSION 1 – AUTHOR RESPONSE

Reviewer: 1

1. Would it be suitable to give a few more information on the organisation of care in England during the study period? Do SCBU have a role in the care for babies born at 27-31 weeks?

This has been addressed in the text on page 4/5 of the marked copy, in Paragraph four of the 'Background and Rationale' and in page 7, 'Methods and Analysis' under Workstream 1 and 3 Population subsection'

2a. It could also be interesting to know the number of networks available in England during the study period

This has been addressed in page 4 paragraph 1 in the 'Background and Rationale'

b. and consequently how the 2 networks will be chosen for parents and clinicians interviews (page 12, line 34).

This has been addressed under Workstream 4 subheading on page 13 paragraph 2

3. Could the authors precise if a special attention will be paid to morbidities or care that could be as important for babies born at 27-31 weeks than BPD, ROP or cranial abnormalities, such as blood stream infections, growth during hospitalisation, breastfeeding initiation and/or at discharge for example? (Page 7, line 13 and 34)

This has been addressed in Paragraph 2, under Clinical outcomes, on Page 8 of the marked document.

addressed

4. "Inclusion and exclusion criteria: units will be offered an opt-out option" (Page 7, line 34). Is it planned to have a minimal dataset to describe units choosing an opt-out option?

This has been addressed on Page 7 under 'Inclusion and exclusion criteria'

5. Is it possible to give a minimal list of "national neonatal auditable standards" that will be taken into account and considered as useful for the care of babies born between 27-31 weeks? (Page 11, line26)

This has been addressed on Page 11/12 under Workstream 2

Personal suggestion: This has been addressed in point 3 and is gratefully accepted

Reviewer 2

1. Page 4, paragraph starting line 52:

“...most pregnant women book for antenatal care and delivery in the hospital closest to their home” ...

This has been addressed in multiple places in the revised document:

paragraph 4 page 4, paragraph 1 page 5, paragraph 1 page 7,

References on how women book in England, the choose and book system has also been added, which has changed the reference numbering for the article. This has been addressed.

2. Page 5, line 43:

Should read “...from BLISS, the charity for babies...”

This has been changed

3. Page 8, line 40 onwards:

Creating a matched sample is a logical step, however it is obviously difficult to assess how large this sample size will be at this stage in the project. As such it may be beneficial to mention that the size of this matched sample may affect how relevant it is to the population generally.

This has been addressed in Paragraph 1 page 9

4. Page 11, line 9:

Should read “...framework. Current thresholds of willingness...”

This has been changed.

Sincerely

Dr T Pillay

Chief Investigator, Opti-Prem

PS I have ensured citations for figures are correct, and the data availability statements now match